# A distributive peptide cyclase processes multiple microviridin core peptides within a single polypeptide substrate

Yi Zhang[1], Kunhua Li [2], Guang Yang[1], Joshua L. McBride[1], Steven D. Bruner[2] & Yousong Ding [1]

Ribosomally synthesized and post-translationally modified peptides (RiPPs) are an important family of natural products. Their biosynthesis follows a common scheme in which the leader peptide of a precursor peptide guides the modifications of a single core peptide. Here we describe biochemical studies of the processing of multiple core peptides within a precursor peptide, rare in RiPP biosynthesis. In a cyanobacterial microviridin pathway, an ATP-grasp ligase, AMdnC, installs up to two macrolactones on each of the three core peptides within AMdnA. The enzyme catalysis occurs in a distributive fashion and follows an unstrict *N*-to-*C* overall directionality, but a strict order in macrolactonizing each core peptide. Furthermore, AMdnC is catalytically versatile to process unnatural substrates carrying one to four core peptides, and kinetic studies provide insights into its catalytic properties. Collectively, our results reveal a distinct biosynthetic logic of RiPPs, opening up the possibility of modular production via synthetic biology approaches.

[1] Department of Medicinal Chemistry, Center for Natural Products, Drug Discovery and Development, College of Pharmacy, University of Florida, Gainesville, Florida 32610, USA. [2] Department of Chemistry, University of Florida, Gainesville, Florida 32611, USA. Correspondence and requests for materials should be addressed to Y.D. (email: yding@cop.ufl.edu)

Ribosomally synthesized and post-translationally modified peptides (RiPPs) constitute a major class of natural products that are increasingly recognized for their biotechnological and biomedical applications[1]. A growing list of diverse post-translational modifications (e.g., macrocyclizations, heterocyclization, prenylation, etc.)[2] allows broad chemical diversity of RiPPs, at a low genetic cost comparable to nonribosomal peptides, a major family of peptidic secondary metabolites produced through modular biosynthesis[3]. Microviridins are a unique family of RiPPs featuring a tricyclic cage-like architecture and possessing potent inhibitory activities toward trypsin-type serine proteases[4–8]. Dysregulation of serine proteases plays a prominent role in the development of many diseases such as cancers, type 2 diabetes, pulmonary disease, Alzheimer's disease, and infectious diseases[9,10]. Microviridins offer a novel scaffold with therapeutic potential. Sixteen microviridin analogs have been isolated from freshwater cyanobacteria and showed substantial chemical variations[4–8]. Recent environmental sampling[11] and bioinformatics analysis[12–14] reveal widespread occurrence of microviridin-related gene clusters even beyond the phylum of cyanobacteria, indicating impressive structural and biosynthetic diversity of microviridins awaiting the discovery. Characterizing and harnessing the biosynthetic logic of these gene clusters could lead to the production of novel microviridin analogs for biomedical applications[15].

The biosynthesis of microviridins is initiated by two ATP-grasp ligases (e.g., MdnC and MdnB). They sequentially form two macrolactones and one macrolactam bond between sidechains of conserved residues of the core peptide (**T**X1**K**YP**S**D**X2D/EE/D**) within the polypeptide precursor MdnA (Fig. 1)[12,16–18]. ATP-grasp enzymes typically catalyze inter-molecular peptide ligations mainly in primary metabolic pathways (e.g., glycolysis and purine biosynthesis)[19–21]. Intra-molecular macrocyclization of the microviridin core peptide with both ester and amide linkages is new to this family of enzymes and importantly represents a unique macrocyclization strategy for RiPP biosynthesis[1]. We recently elucidated the structural basis of MdnA/B/C in the microviridin J pathway[16], and revealed a distinct, α-helical leader peptide/processing enzyme interaction. Other processing enzymes in the microviridin biosynthesis include an unidentified protease that cleaves off the processed core peptide, an N-acetyltransferase MdnD and an ABC transporter MdnE. Leveraging the two ATP-grasp ligases, we[16] and the Dittmann group[14,22] partially reconstructed the biosynthesis of microviridin in vitro, which can be a useful approach to produce RiPP analogs with desirable properties[22–25].

Processing multiple core peptides within a single precursor substrate is a rare RiPP biosynthesis strategy that has so far been described only in the production of cyanobacterial cyanobactins[26], plant cyclotides[27] and orbitides[28], and fungal ustiloxins[29] and phomopsins[30]. The current understanding of the underlying biosynthetic logic has completely come from the seminal studies of cyanobactin biosynthetic pathway[25]. A heterocyclase (e.g., TruD) first interacts with the leader peptide primarily through a common precursor peptide recognition element[31], a mechanism shared by the majority of currently known RiPPs classes but not microviridin[16]. Next, this enzyme modifies natural and unnatural precursor peptides likely in a processive manner and following a C-to-N directionality[25,32], although the nature of enzyme substrates can affect enzyme catalytic performance. Despite obvious differences in the nature of biotransformations (e.g., substrate and the type of chemical reactions), the observed processivity and directionality of cyanobactin heterocyclase partially resemble the modular biosynthesis of primary and secondary metabolites fatty acids, polyketides, and nonribosomal peptides, which generally follows the

colinearity rule and releases the biosynthetic intermediates only after the catalysis of the last module[33].

Here we report biochemical characterization of the macrocyclization of a microviridin precursor peptide (AMdnA) carrying three core peptides. We show that AMdnC, a homolog of macrolactone-forming MdnC[16], converts AMdnA into multiple species representing each predicted macrolactonization stage on the three core peptides, and that the processing possesses a unique combination of enzymatic features as the distributive nature and two-level directionality, offering a valuable example for enzymology investigation. Furthermore, we probe the plasticity of the microviridin biosynthesis as the processing of engineered AMdnA substrates carrying one to four core peptides by AMdnC, and kinetic studies provide useful mechanistic insights into the enzyme catalytic properties.

## Results

**AMdnC modifies AMdnA with multiple macrolactone linkages.** Our bioinformatics analysis of publicly available genomic database discovered the microviridin gene cluster mainly from cyanobacteria but also representatives from, for example, Bacteroidetes (e.g., *Microscilla marina*) and Proteobacteria (e.g., *Sorangium cellulosum*) (Supplementary Fig. 1). Intriguingly, the precursor peptide (AMdnA) of one cluster from the filamentous cyanobacterium *Anabaena* sp. PCC7120 contains three predicted core peptides (Fig. 2a), while AMdnC shares over 60% amino acid identity with multiple MdnC homologs (Supplementary Fig. 2), suggesting that this system can likely offer a new glimpse into the mechanism of modular RiPP biosynthesis. We therefore expressed and purified recombinant AMdnA with either an N- or C-terminal His$_6$-tag in *Escherichia coli* (Supplementary Fig. 3). Tag-free AMdnA was produced by the enzymatic removal of the N-His$_6$-tag. Partially purified recombinant AMdnC was obtained from *E. coli* C43 (DE3) culture when coexpressed with a chaperone plasmid pGro7[34] (Supplementary Fig. 3). Substantial efforts[35,36] to remove concomitant proteins from AMdnC achieved no meaningful improvement of its purity.

We next examined the processing of AMdnA by AMdnC. Using high-resolution (HR)-MS analysis, we observed a cluster of species in the reaction, whose molecular weights (MWs) were smaller than AMdnA by repeats of 18 Da (Fig. 2b). The formation of one intra-molecular lactone bond on the parent molecule is reflected as the loss of one water (18 Da, dehydration Δ), and this MS result, therefore, suggests the occurrence of serial enzymatic macrolactonizations on the precursor peptide AMdnA. The most abundant species in the enzyme reaction contained five dehydrations (AMdnA-Δ5). Since up to two macrolactonizations are predicted to take place on one microviridin core peptide, these data suggest the processing of two and half cores by AMdnC. AMdnC generated the same product profile from AMdnA carrying the C-His$_6$-tag (Supplementary Fig. 4) but no activity was observed with the N-His$_6$-tagged substrate. Interestingly, the use of the minimal leader peptide MdnA$_{9-22}$ that activates MdnC in *trans* in our previous work[16] resulted in up to five dehydrations on this inactive substrate (Supplementary Fig. 5). This result suggests that the N-His$_6$-tag negatively affects catalytically critical interactions between the leader peptide of AMdnA and AMdnC[16] and the catalysis of AMdnC can be controlled in *trans*, a feature shared with the processing of the microviridin precursor peptides containing a single core peptide[16,22].

We further optimized the reaction conditions of AMdnC (Supplementary Fig. 6). Under the optimal conditions, one product species with seven dehydrations appeared in the reaction of AMdnC, unexpected as a predicted maximum of six would result from fully processing the three core peptides within

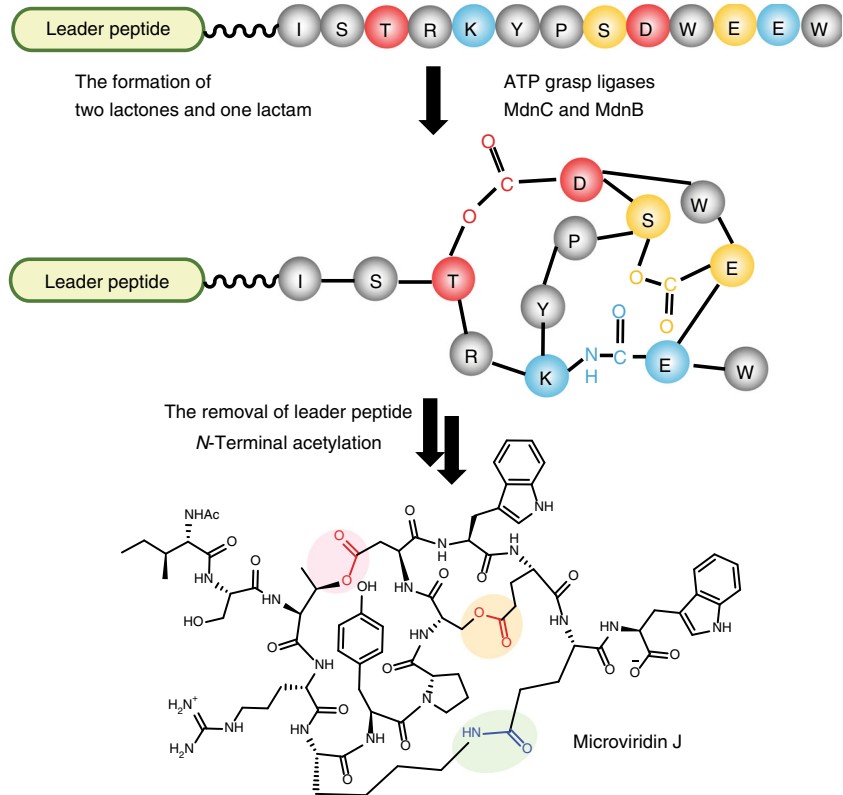

**Fig. 1** Schematic representation of microviridin J biosynthesis. MdnC installs two macrolactones on MdnA, followed by the macrolactamization by MdnB. The modified core is then released for an *N*-acetylation modification

AMdnA (Fig. 2b). Two negative controls with recombinant chaperone GroEL (Supplementary Fig. 3) and boiled AMdnC did not produce any detectable amount of processed AMdnAs under the same reaction conditions (Supplementary Fig. 7). To further probe the catalytic activity of AMdnC, we mutated its K165 and D280 with alanine, which are predicted to interact with $Mg^{2+}$ and ATP, respectively[16], and are strictly conserved among MdnC homologs (Supplementary Fig. 2). The two recombinant AMdnC mutants had the similar level of impurity to wild type (Supplementary Fig. 3) and were catalytically inactive toward AMdnA (Supplementary Fig. 8). This result suggested the essentiality of $Mg^{2+}$ and ATP to the AMdnC reaction, the same as other ATP-grasp ligases, and iterated that the formation of detected dehydration species in the AMdnC reaction is not driven by any impurity present in the enzyme solution (Supplementary Fig. 3).

**Structural determination of the processed AMdnA**. We developed a proteomic approach involving GluC endoprotease digestion to structurally characterize the major processed AMdnA species in the AMdnC reaction. GluC selectively and effectively cleaves peptide bonds *C*-terminal to glutamic acid residues and theoretically would cleave off each of the three core peptides within AMdnA (Supplementary Fig. 9a). LC-HR-MS analysis of the digestion mixture of intact AMdnA substrate identified three fragments (F1-3) carrying the core peptide M1, M2, and M3, respectively (Fig. 2c). We also observed one fragment containing both M1 and M2 (F12, Fig. 2c). By contrast, both F1 and F12 almost completely disappeared from the LC trace of the digested mixture of processed AMdnA, and two new major peaks with retention times of about 8.0 min and 8.7 min emerged (Fig. 2c). HR-MS analysis assigned these two new peaks as F2-Δ2 and F3-

Δ2, respectively, and also identified two small peaks with retention times of 6.6 min and 7.5 min as F1-Δ1 and F12-Δ3, respectively (Fig. 2c, Supplementary Fig. 9). Therefore, AMdnC catalyzed the corresponding loss of one, two and two waters from the M1, M2, and M3 within AMdnA as the major processed species. The observed dehydration pattern suggests a 1-2-2 ring topology of the major species as shown in Fig. 2d. We further validated this deduced structure by comparative MS/MS analysis (Fig. 3). In tandem MS traces, multiple fragments were generated from the M2 and M3 of the F2 and F3 (Fig. 3a, c). Expectedly, these fragments disappeared in the traces of the F2-Δ2 and F3-Δ2 because the formation of two macrolactones blocked the fragmentation of the core peptide (Fig. 3b, d). The large sizes of the F1 and F1-Δ1 led to unanalyzable MS/MS fragmentation as observed in other reports[37]. Finally, we quantitated the number of lactone bonds in the major species by using the reducing agent LiBH4[13]. After reduction, one lactone bond is converted into two −OH groups, with a diagnostic mass increase of 4 Da. A 20-Da increase was observed upon the treatment of the reductant, indicating the existence of five lactones in the major species of the AMdnC reaction (Supplementary Fig. 10)[13].

**AMdnC processes AMdnA in a distributive fashion**. The formation of multiple species during variable stages of macrolactonization steps indicates a distributive nature of AMdnC in processing AMdnA (Fig. 2b). To further investigate this catalytic feature, we lowered the molar ratio of substrate/enzyme to 115:1 and then monitored the reaction course from 0 to 16 h (Fig. 4). After 0.5 h, AMdnC transformed AMdnA into the species with only one or two dehydrations, while the new species with three to five dehydrations appeared after 2 h. Further extending the reaction time to 16 h led to a modest shift of the product profile

with the formation of the species with the loss of six waters (Fig. 4). The formation of intermediates with all possible dehydration events over time indicates a distributive catalysis of AMdnC, suggesting AMdnC binds intact AMdnA, forms a first macrolactone, dissociates from the processed intermediate, and then recaptures it as well as intact AMdnA as the substrates for subsequent macrolactonizations. This process is iterative toward the formation of more advanced species (Fig. 4). In the RiPP biosynthesis, only few enzymes, which typically process the RiPP precursor peptides containing a single core peptide, possess the distributive catalytic property, e.g., LctM involved in lantipeptide biosynthesis[38], OphA for the methylation of omphalotin[39,40] and PsnB for the cyclization of four repeated units of plesiocin[41]. On the other hand, TruD demonstrated the processive catalysis when the unnatural substrate with a single cyanobactin core peptide was used as the substrate[32].

**The directionality of AMdnC in processing AMdnA.** The distributive nature of catalysis by AMdnC results in the accumulation of intermediates at each stage of substrate processing, and offers the opportunity to investigate the directionality of the catalysis, an observed aspect of modular RiPP biosynthesis[25]. To probe this feature, we terminated the reactions after 0, 0.5, 2, and 16 h (Fig. 4) and then determined the location of lactone bonds of processed species by GluC-based proteomic analysis. At 0.5 h, we observed peaks for F1-Δ1, F2-Δ2, and F3-Δ1 (Fig. 5), which collectively constitute the intermediates with the loss of one and two waters as shown in Fig. 4. This result suggests that AMdnC is able to macrolactonize any of three core peptides within AMdnA as the initial step. On the other hand, the F1 signal decreased to a greater extent than F2 and F3, indicating that AMdnC favors the M1 in the processing (Fig. 5). In line with this observation, the signal ratio of F1-Δ1/F1 increased along the reaction course, and was significantly higher than that of the F2-Δ2/F2 at all time points (Fig. 5). Furthermore, the majority of the M1 and M2 were processed after 16 h, while a substantial amount of F3 remained intact. At 16 h, we observed only F3-Δ2, whose retention time was the same as F3-Δ1 in the LC-HRMS analysis. These results demonstrate an unstrict but favored N-to-C directionality of AMdnC in processing three core peptides within AMdnA. In comparison, LctM and OphA with the distributive catalytic feature show the unidirectionality, while TruD strictly follows the C-to-N directionality in maturing the unnatural substrate carrying a single cyanobactin core peptide[32]. Unidirectional catalysis is also the prevailing feature of modular biosynthesis of primary and secondary metabolites[42,43]. In this regard, AMdnC represents a highly valuable example for the investigation of processing directionality in natural product biosynthesis.

In addition to the directionality in the processing of multiple core peptides, individual core peptides within AMdnA are macrocyclized by two macrolactones, therefore demonstrating topological directionality. In an early report, MdnA is modified first with the larger and then the smaller macrolactone (Fig. 1)[44]. To investigate this aspect in the processing of AMdnA, we created six AMdnA alanine mutants by individually mutating each acidic residue required for the formation of the first (D51, D76, and D93) or the second (E53, E78, and D95) macrolactone (Fig. 2d, Supplementary Figs. 3 and 11). Both D76A and D93A mutants lost up to three waters in the AMdnC reaction as shown in the LC-HR-MS analysis (Fig. 6a), indicating that these mutations block the formation of the first and subsequently the second macrolactone on the M2 and M3, respectively (Supplementary Fig. 11c and 11e). By contrast, only the second macrolactone was not formed on the M1, M2, and M3 in the corresponding E53A, E78A, and D95A mutants (Supplementary Fig. 11b, 11d, and 11f),

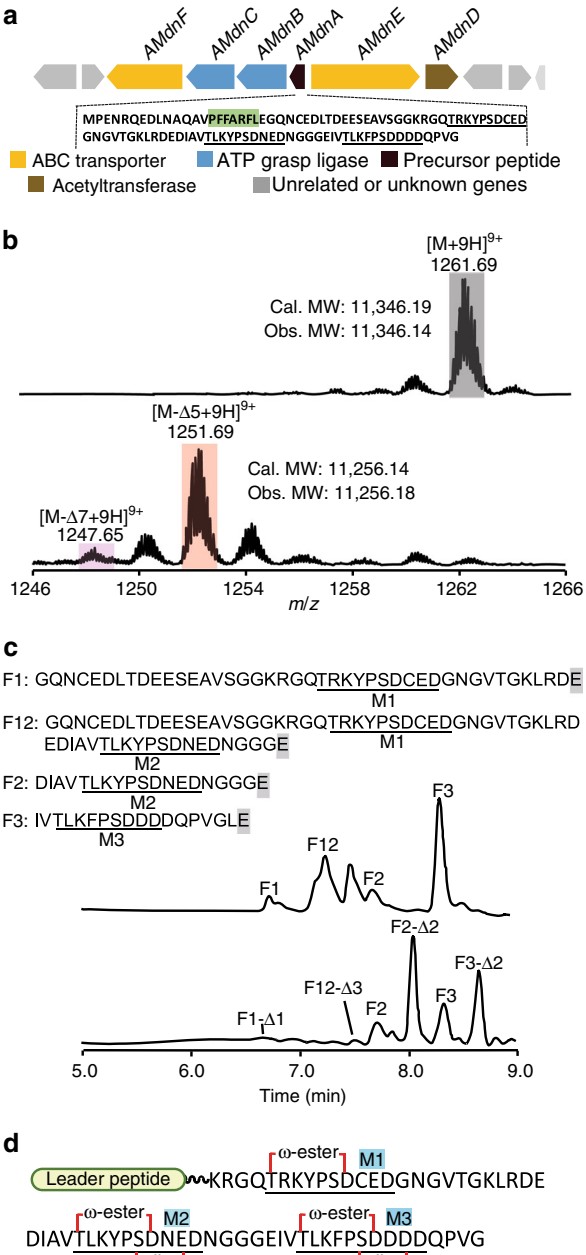

**Fig. 2** AMdnC macrolactonizes the three core peptides within AMdnA. **a** A putative microviridin gene cluster was identified from *Anabaena* sp. PCC7120. The conserved binding motif of the leader peptide within AMdnA is shaded in green while three predicted core peptides are underlined with solid lines. **b** HR-MS analysis detected an array of species with one to seven dehydrations (Δ) after incubating AMdnA with AMdnC at a molar ratio of 45:1 for 16 h. The most abundant species lost five waters from AMdnA. **c** HPLC traces of intact (up) and processed (down) AMdnAs after GluC digestion. Key chromatographic peaks were labeled with the names of corresponding peptide fragments released by GluC. **d** The deduced structure of the most abundant species in the AMdnC reaction. Key acidic residues involved in the lactonizations of core peptides were numbered

which lost up to five, four, and four waters, respectively (Fig. 6b). We further confirmed the ring topology of the major products in these reactions by the GluC-based proteomic analysis (Supplementary Fig. 12). These results show that AMdnC follows a strict order as sequentially forming the larger and smaller macrolactones on each core peptide within AMdnA and the formation

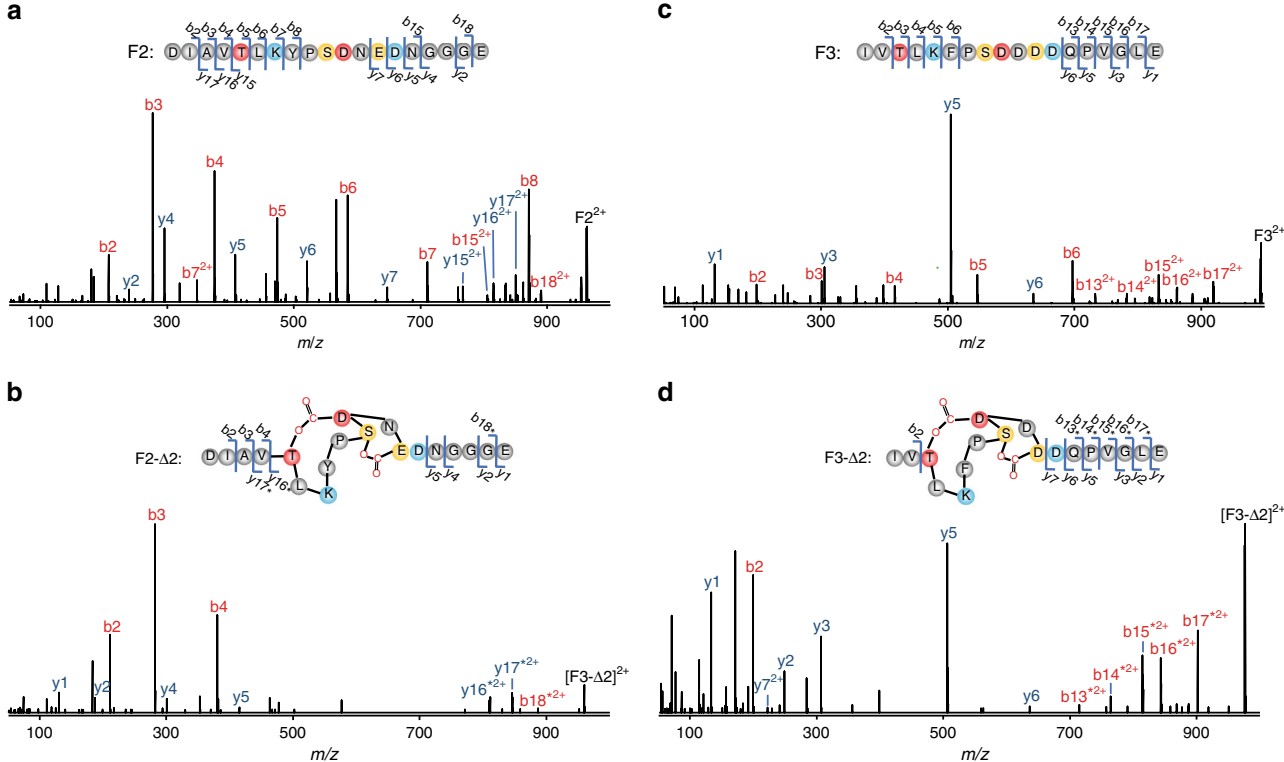

**Fig. 3** Structural determination of AMdnA-Δ5 by HR tandem MS analysis. **a** HR-MS/MS spectrum of the released F2 of intact AMdnA. **b** HR-MS/MS spectrum of the released F2-Δ2 of processed AMdnA. **c** HR-MS/MS spectrum of the released F3 of intact AMdnA. **d** HR-MS/MS spectrum of the released F3-Δ2 of processed AMdnA. The samples were treated with GluC for 16 h. Macrolactonizations on the M2 and M3 prevented the formation of multiple fragments from the corresponding regions of F2-Δ2 and F3-Δ2

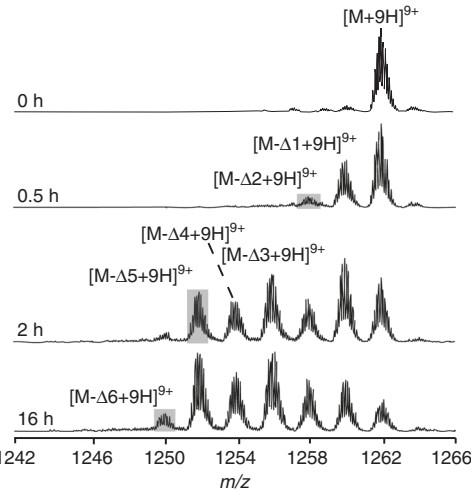

**Fig. 4** AMdnC processes AMdnA in a distributive manner. HR-MS analysis showed the formation of intermediates with varying degrees of dehydrations after incubating AMdnA with AMdnC for 0, 0.5, 2, and 16 h. The molar ratio of substrate/enzyme was set to 115:1 to assist the detection of accumulated intermediates

of the second macrolactone depends on the occurrence of the first one. We observed up to five dehydrations, instead of the predicted maximum of four, from the D51A mutant in the AMdnC reaction (Fig. 6a). Further analysis of the AMdnA sequence led us to identify a 'quasi core peptide' from the inter-modular region of the M1 and M2 (Fig. 2a, d). The sequence of the quasi core (**T**GK**L**RD**E**) has significant deviations from that of the full core peptide (**T**X1**K**YP**SD**X2**D/EE/D**) but can

presumably form one macrolactone between its Thr and Glu residues (Supplementary Fig. 11a). Evidently, GluC digestion of the processed AMdnA D51A identified F12-Δ3, F2-Δ2, and F3-Δ2, but not F1-Δ1 that was found from the digested processed AMdnA (Supplementary Fig. 12a). The macrolactonization of the quasi core can make the Glu residue inaccessible to GluC for releasing the F1-Δ1. The formation of one macrolactone on the quasi core can account for the loss of three waters to generate F12-Δ3 from the D51A (Supplementary Fig. 12a) and lead to the formation of the species with seven dehydrations from AMdnA (Fig. 2b, Supplementary Fig. 4). Collectively, these results reveal the topological directionality of AMdnC in processing individual core peptides of AMdnA, demonstrate the independent processing of each core peptide, and also suggest no formation of the macrolactones across different core peptides. Importantly, the formation of multiple intermediates from the AMdnA mutants (Fig. 6) provides new lines of evidence indicating both the distributive nature and the unstrict overall directionality of AMdnC catalysis.

**The processing of engineered AMdnA substrates by AMdnC.** The catalytic versatility of AMdnC toward AMdnA alanine mutants encouraged us to further assess its substrate scope. We first used the natural microviridin precursor peptide MdnA carrying the single core peptide and observed the MdnA-Δ2 as the only product of the AMdnC reaction as shown in LC-HR-MS analysis (Fig. 7a). In comparison, MdnC is known to install two macrolactones on MdnA[16]. In this work, we found that MdnC generated the AMdnA-Δ1 as the major product and a small amount of AMdnA-Δ2, but no other advanced species (Fig. 7b). These results suggest the superior catalytic versatility of AMdnC. To further investigate the promiscuity of AMdnC, we prepared

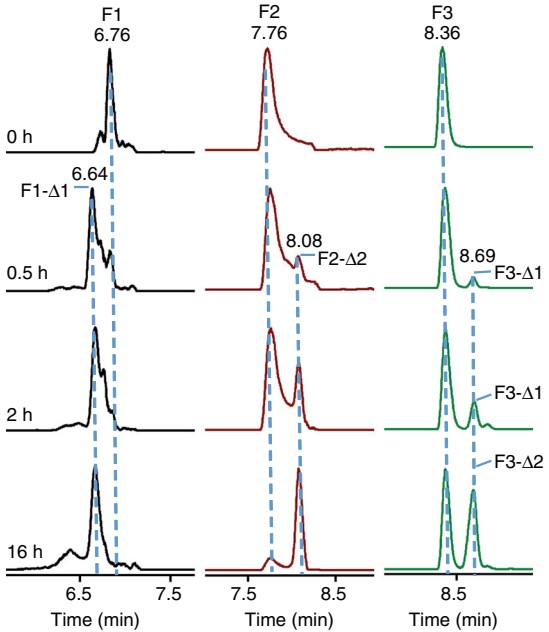

**Fig. 5** Overall directionality of AMdnC in processing AMdnA. Extracted ion chromatograms of the fragments of intact and processed AMdnAs at different time points revealed the unstrict but favored *N*-to-*C* overall directionality of AMdnC. F3-Δ1 and F3-Δ2 had the same retention times in LC-HR-MS analysis

six engineered AMdnA variants containing one to four core peptides, M1V1, M1V2, M1M2, M2M3V1, M2M3V2, and M123M3 (Fig. 7c, Supplementary Fig. 3 and Supplementary Table 2). Here, M1, M2 and M3 represented the first, second, and third core peptide of AMdnA (Fig. 2d), while V1 and V2 indicated the presence and absence of the quasi core peptide, respectively. LC-HR-MS analysis revealed that AMdnC successfully processed the unnatural substrates carrying one and two core peptides, M1V1, M1M2, and M2M3V1 (Fig. 7d–f), and promoted up to eight dehydrations on the M123M3 (Fig. 7g), indicating the processing of the fourth core peptide. We did not observe any processed M1V2 and M2M3V2 species (Supplementary Fig. 13), suggesting the quasi core peptide of AMdnA might influence the catalysis of AMdnC in an unclear manner. Nonetheless, the results demonstrate the exceptional substrate promiscuity of AMdnC and lay the basis for the further investigation of the evolution of microviridin biosynthesis.

**Kinetic characterization of AMdnC.** To provide additional mechanistic insights into the catalysis of AMdnC, we performed kinetic studies. The distributive catalysis of AMdnC compounded by the unstrict reaction directionality leads to a number of potential processed intermediates (i.e., there is a maximum of 9 possible AMdnA-Δ3 species). Determining the kinetic constants of each species using LC-MS analysis is a complex problem (each likely has varying $K_m$ and $k_{cat}$) and technically challenging, due to its low concentration in the reaction and the lack of multiple suitable internal standards. Instead, we sought to measure the gross reaction kinetics. Specifically, we employed the HPLC analysis to accurately quantitate the net production of ADP from ATP that accounts for all phosphorylation reactions by AMdnC, which lead to subsequent macrocyclizations (Supplementary Fig. 14a). Control reactions that were the same as enzymatic reactions except no substrate were used to subtract the ADP produced by automatic ATP hydrolysis or any ATP using

enzymes co-purified with AMdnC. This approach determined a measured $K_m$ (AMdnA) of $26.9 \pm 0.2\,\mu M$ in the AMdnC reaction (Table 1, Supplementary Figs. 14b and 15a), comparable with MdnA ($K_m = 23.8 \pm 1.2\,\mu M$) that was determined by quantitating the concentrations of MdnA-Δ2 in the MdnC reaction. On the other hand, the apparent $k_{cat}$ (ATP, $12.4 \pm 1.4\,min^{-1}$) of AMdnC was about 25-times higher than MdnC (MdnA, $0.47 \pm 0.02\,min^{-1}$). To further understand the catalytic performance of AMdnC, we included varying concentrations of MdnA and M1V1 in the kinetic analysis. Interestingly, we found that AMdnC displayed an approximate two-fold higher efficiency ($k_{cat}/K_m$) toward these two unnatural substrates with a single core peptide than AMdnA while maintaining a similar $K_m$ value (Table 1, Supplementary Fig. 15b, c). To examine how the approach (the net production of ADP) used to determine the AMdnC kinetics affected the measured kinetic constants, we used the HPLC analysis to quantitate the concentrations of MdnA-Δ2 in the AMdnC reaction with MdnA as the substrate (Fig. 7a). We found that the $K_m$ values determined by these two approaches were at the same level ($24.4 \pm 0.8\,\mu M$ vs $28.1 \pm 1.0\,\mu M$) while the apparent $k_{cat}$ (MdnA, $0.7 \pm 0.1\,min^{-1}$) was 39 times lower than that (ATP, $27.4 \pm 0.2\,min^{-1}$) of the same reaction when determined on the basis of ADP production (Table 1, Supplementary Fig. 15d). The significant difference of two apparent $k_{cat}$ values of the same reaction suggested that the final macrocyclization is likely the rate-limiting step of AMdnC reaction. Importantly, the same level of measured $K_m$ values with AMdnA, MdnA and M1V1 as substrates indicated that the leader peptide of precursor peptide substrates is primarily responsible for the catalytically determining interactions with AMdnC. Indeed, the binding affinities ($K_D$) of AMdnA and M1V1 with AMdnC were 2.7 and 2.3 μM, respectively, at the same level (Supplementary Fig. 16a, b). Furthermore, we created a quadra AMdnA mutant (AMdnAi) carrying four alanine mutations D51A, T59A, D76A, and D93A, which are expected to correspondingly abolish the macrolactonization on the M1, quasi core, M2 and M3. Compared with AMdnA and M1V1, this inactive analog possessed a slightly higher binding affinity ($K_D = 3.6\,\mu M$) with AMdnC (Supplementary Fig. 16c). Expectedly, no macrocyclized species was produced from AMdnAi by AMdnC and the net production of ADP was at the same level of the negative control (Supplementary Fig. 17a, b). However, we observed that AMdnAi inhibited the enzymatic processing of AMdnA in a dose-dependent manner (Supplementary Fig. 17c). Only AMdnA-Δ1 was produced from AMdnA when 3.8 μM of AMdnAi were included in the reaction. Kinetic analysis further revealed AMdnAi to be a competitive inhibitor as it increased the $K_m$ values of AMdnA while the changes of observed $k_{cat}$ values were not statistically significant (Table 1). Therefore, AMdnAi competes with AMdnA and processed species in the interactions with AMdnC. As such, it inhibits the distributive catalysis of AMdnC with a calculated $K_i$ of $1.3 \pm 0.2\,\mu M$ (Supplementary Fig. 17d). Furthermore, our kinetic studies reveal that the presence of multiple core peptides within AMdnA does not enhance the rate of enzyme-catalyzed ATP hydrolysis (Table 1). For processive catalysis, the concatenation of multiple modification sites within a single substrate could increase the catalytic efficiency as reducing the search dimension of enzyme to one (*N* to *C* or *C* to *N*), as shown with the restriction enzyme *Eco*RI in a previous report[45].

## Discussions

Multiple modifications on the substrate by the same enzyme can occur in two alternative catalytic modes, processivity or distributivity[46,47]. Processive catalysts associate with their substrates and then perform multiple rounds of reactions until the

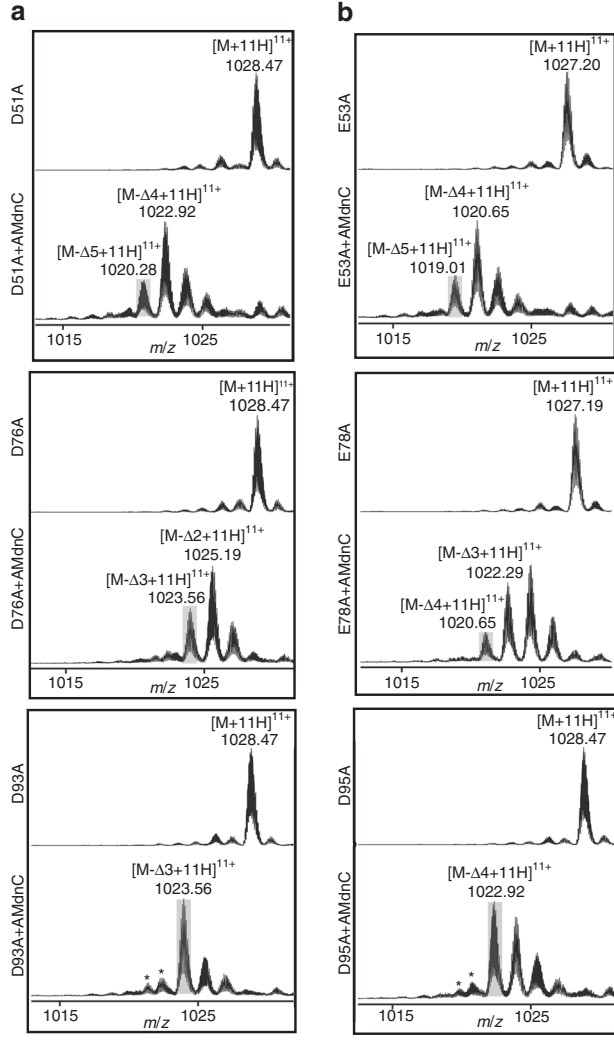

**Fig. 6** Topological directionality of AMdnC in processing individual core peptides of AMdnA. **a** Three key acidic residues required for the formation of the first lactone on each core peptide within AMdnA were individually substituted with alanine. The resultants were used as the substrates of AMdnC. **b** Three key acidic residues involved in the formation of the second lactone were individually substituted with alanine to create AMdnA mutants as the substrates of AMdnC. The most processed intermediates were shaded in gray. Unrelated minor MS peaks were labeled with *

formation of final products, while a distributive catalyst dissociates from the processed intermediates after each transformation. In comparison, the processive catalysis increases the effective molarity of the substrate by enforcing the one-dimension movement of the catalyst (e.g., either $N$ to $C$ or $C$ to $N$ directionality on peptidic substrates), while the free diffusion in three-dimensional space is associated with distributive enzymes such as GCN5 that acetylate multiple lysine residues on a single histone[48] and glycogen synthase kinase-3 in phosphorylating microtubule-associated tau protein[49]. Accommodatingly, processive catalysis requires a nonspecific substrate recognition, which allows unidirectional sliding of catalyst. Using DNA polymerase as an example, the binding is largely mediated by electrostatic interactions between the enzyme and the phosphate backbone and minor groove of the DNA template[50]. In the microviridin biosynthesis, we revealed the specific binding between the α-helical region of the leader peptide and the β9β10 loop of MdnC, which allosterically activates the enzyme[16]. This featured interaction is likely preserved for the catalysis of AMdnC, since a homology

modeled structure, including the β9β10 loop, highly resembles MdnC (PDB code 5IG9, Supplementary Fig. 18). Furthermore, the results of serial experiments in this work supported the determining role of the leader peptide for the enzymatic processing of core peptide, for example, the MdnA$_{9-22}$ activated AMdnC to process an otherwise-inactive AMdnA with the $N$-His$_6$ tag (Supplementary Fig. 5); AMdnC was able to process MdnA and engineered AMdnA variants (Fig. 7); and AMdnAi competitively inhibited the catalysis of AMdnC (Supplementary Fig. 16). On the other hand, the leader peptide/enzyme interaction may have a mere contribution to the distributive catalytic mode of AMdnC; obviously, the MdnA$_{9-22}$ is unable to act as a hinge to constrain the sliding of the enzyme on the $N$-His$_6$ tagged AMdnA to generate multiple processed species (Supplementary Fig. 5).

The directionality in processing a single core peptide has been elucidated in several RiPP classes, such as thiazole/oxazole-modified microcins (TOMMs)[51] and lantipeptide[38,52,53]. The $N$ to $C$[38,54,55], $C$ to $N$[51,52], and nonlinear but still-ordered processings[53] all have been reported. In this work, AMdnC demonstrated a two-level directionality. The formation of two lactones on individual cores within AMdnA follows a stringent order (Fig. 6), comparable to the $N$ to $C$ directionality of single core peptide systems, while processing the three core peptides within AMdnA by AMdnC showed an unstrict directionality (Fig. 5). As a comparison, the unidirectionality was observed in the modification of the unnatural substrate with a single cyanobactin core by TruD[32] and in the modular biosynthesis of fatty acids, polyketides, and nonribosomal peptides[33]. The directionality feature of AMdnC may be linked with its distributive catalysis. On the other hand, whether and how other biosynthetic enzymes (e.g., AMdnB) may influence the macrolactonizations of AMdnA is unknown.

Distributive catalysis has been reported with several RiPPs processing enzymes including microcin B17 synthetases[54], lantipeptide processing enzymes NisB[55,56], LctM, HalM2[38], and LabKC[52], OphA for the $N$-methylation of omphalotin[39], and recently characterized ATP-grasp enzyme PsnB[41]. These enzymes generally process the substrates that carry only a single core peptide, and display a certain degree of directionality[57]. Biochemical studies with unnatural and natural cyanobactin precursor peptides suggest that the heterocyclases can be processive and unidirectional[32] and the catalytic properties can be influenced significantly by the substrates of the reactions[58,59]. Compared with these enzymes, AMdnC distributively processed the substrate carrying the three core peptides and possessed the two-level directionality. These features suggest AMdnC as a useful example for advancing the understanding of RiPPs processing enzymes in general.

In this work, we determined catalytic kinetic parameters of AMdnC by quantitating the net production of ADP in the reaction. This approach circumvented the technical difficulty to directly measure the concentrations of each processed species in LC-MS analysis, particularly when AMdnA was used as the substrate, and has previously been used to determine the steady state kinetics of TruD[59]. One assumption of this approach, however, is that the ATP hydrolysis is the rate-limiting step of AMdnC reaction. Our studies revealed that the calculated $K_m$ values of AMdnA, M1V1, and MdnA by this approach were at the same level as those of MdnA in the MdnC and AMdnC reactions (Table 1), which were determined by quantitating the MdnA-Δ2 concentrations by HPLC analysis in the current and our previous work[16]. On the other hand, the apparent rates ($k_{cat}$) of ATP hydrolysis in the AMdnC reactions were 20–60 times faster than those determined by the HPLC method (apparent $k_{cat}$ values varied from 0.47 min$^{-1}$ to 0.7 min$^{-1}$)[16]. These results

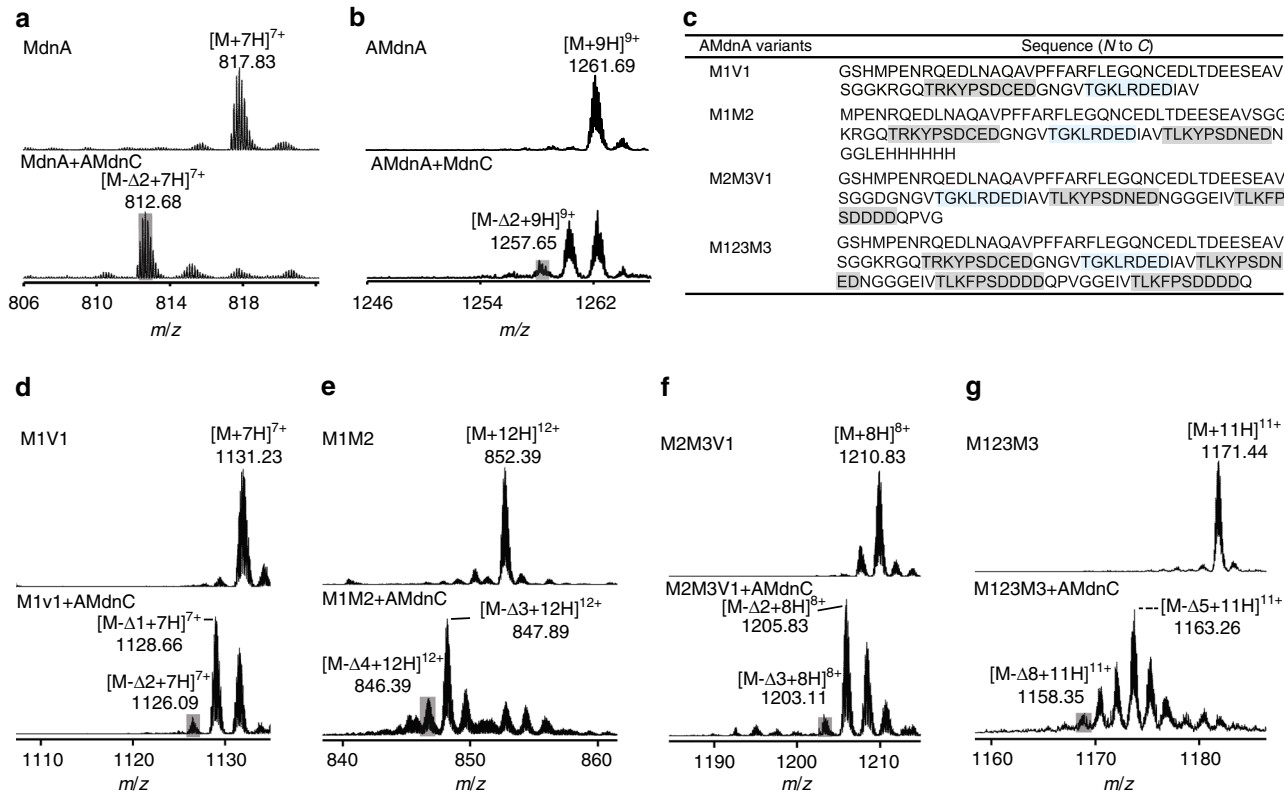

**Fig. 7** HR-MS analysis showed the processing of natural and unnatural microviridin precursor peptides. **a** AMdnC introduced two lactones on MdnA, the precursor peptide of microviridin J. **b** MdnC installed up to two lactones on AMdnA. **c** Sequences of select AMdnA variants. The core peptides and quasi core peptide were shaded in gray and light cyan, respectively. **d** Processed M1V1, **e** processed M1M2, **f** processed M2M3V1, and **g** processed M123M3 in the AMdnC reaction were shown in HR-MS spectra. The most processed intermediates were shaded in gray

**Table 1 Kinetic parameters of AMdnC in processing AMdnA, MdnA and M1V1[a]**

| Substrate analogs | $K_m$ (µM) | $k_{cat}$ (min$^{-1}$) | $k_{cat}/K_m$ (min$^{-1}$µM$^{-1}$) |
|---|---|---|---|
| M1V1 | 22.0 ± 0.2 | 20.8 ± 0.1 | 0.95 ± 0.01 |
| MdnA | 24.4 ± 0.8 | 27.4 ± 0.2 | 1.1 ± 0.04 |
| MdnA[b] | 28.1 ± 1.0 | 0.70 ± 0.1 | 0.03 ± 0.001 |
| AMdnA | 26.9 ± 1.4 | 12.4 ± 0.7 | 0.46 ± 0.07 |
| AMdnA[c] | 38.3 ± 1.3 | 13.6 ± 0.1 | 0.34 ± 0.01 |
| AMdnA[d] | 42.8 ± 2.5 | 12.5 ± 0.4 | 0.28 ± 0.02 |
| AMdnA[e] | 55.5 ± 3.0 | 11.7 ± 0.3 | 0.21 ± 0.01 |
| AMdnA[f] | 87.9 ± 6.1 | 12.9 ± 0.5 | 0.15 ± 0.01 |

[a]Data represented mean ± s.d. ($n \geq 3$)
[b]kinetic parameters were determined by HPLC-based quantitation of MdnA-Δ2 in the AMdnC reaction
[c-f]AMdnC reactions contained 0.55 µM, 1.1 µM, 1.9 µM, and 3.8 µM AMdnAi, respectively

suggest that the stability of substrate-enzyme complex is primarily determined by the interactions of the lead peptide/cyclase but the macrocyclization is likely the rate-limiting step, which involves the nucleophilic attack of the phosphorylated carboxylic acid by the –OH group (Supplementary Figure 14a). Compared with TruD ($K_m$ = ~1 µM, $k_{cat}$ for ATP hydrolysis = 2.6 min$^{-1}$)[59], AMdnC showed an around 25-times less tight interaction with its substrate, but about 10-fold higher rate of ATP hydrolysis. The relatively low stability of the substrate-AMdnC complex can be beneficial to the distributive catalysis of multiple core peptides. Compared with one glutathione synthetase from *Arabidopsis thaliana* [$K_m$ (γEC) = 39 ± 5 µM, $k_{cat}$ (ATP) = 12.2 ± 0.3 s$^{-1}$, based on the net production of ADP][60], an ATP-grasp ligase of

the primary metabolism, the catalytic efficiency of AMdnC was about 20 times lower, consistent with the previous finding that enzymes of the secondary metabolism are typically ~30-fold slower than their counterparts of primary pathways[61].

Collectively, this work indicates that AMdnC is catalytically active toward AMdnA. We reveal that the macrolactonizations of three core peptides within AMdnA requires continuous cycles of substrate binding and release and follows the two-level directionality. The combination of these two catalytic properties represents a valuable example to the better understanding of biosynthetic enzymes of primary and secondary metabolites. Our work further offers new information about the modular RiPPs biosynthesis, sheds light into the evolution of microviridin biosynthetic systems, and provides a framework for future synthetic biology efforts to produce microviridin analogs as serine protease inhibitors.

## Methods

**Construction of expression plasmids of AMdnC and mutants**. The *AMdnC* gene (Genbank: NC_003276.1 REGION: 7321-8292) was amplified from isolated genomic DNA from *Anabaena* sp. PCC7120 in a PCR reaction using primers AMdnC-FW and AMdnC-RV (Supplementary Table 1). The PCR product was cloned into pET28a to yield the expression construct pET28a-AMdnC following standard molecular biology protocols[62]. The insert in the construct was sequenced to exclude errors. To create AMdnC mutants, the pET28a-AMdnC was used as the template in site-directed mutagenesis PCR reactions with primers shown in Supplementary Table 1.

**Construction of expression plasmids of AMdnA and mutants**. The *AMdnA* (NC_003276.1 REGION: 9429-9731) gene was amplified as described above. Plasmids pET28a and pET30b were used for expressing AMdnA with *N*- and *C*-His$_6$ tag, respectively. To create AMdnA mutants, the pET28a-AMdnA was used as the template in site-directed mutagenesis PCR reactions with primers shown in Supplementary Table 1. To create AMdnA variants, the AMdnA gene was used as

the template in the PCR reactions using primers shown in Supplementary Table 1. More specifically, the fragments of M1V1, M1V2, M1M2, and M123M3 were amplified using the primer AMdnA-FW and the corresponding reverse primers. To prepare M2M3V1 and M2M3V2, fragments of the leader peptide and core peptide domains were separately amplified from the AMdnA gene and then fused in overlapping PCR reactions. All final inserts were sequenced to exclude errors and cloned as described above.

**In vitro reconstruction of the AMdnC reaction.** The AMdnC reaction was carried out in the mixture (100 µL) containing 40 µM AMdnA, 50 mM buffer, 2 mM ATP, 10 mM divalent cation, and 50 mM KCl. Optimization of reaction conditions included buffer systems (Tris, phosphate, MOPS, and HEPES), pH (4–9), divalent cations ($Co^{2+}$, $Fe^{2+}$, $Ca^{2+}$, $Cu^{2+}$, $Zn^{2+}$, and $Mg^{2+}$), additives (5% glycerol, 3 mM BME, and 3 mM dithiothreitol), and enzyme doses (0.35 µM, 0.90 µM, and 1.6 µM), which were performed in a sequential and cumulative manner. The reactions were incubated at 37 °C for 16 h and terminated by an equal volume of methanol. The resulting mixtures were centrifuged (14,000× g, 20 min, 4 °C) and 50 µL of supernatants were used for HPLC and LC-MS analysis, whose parameters were detailed below. The optimal reaction mixture contained 50 mM HEPES, pH 8.0, 5% glycerol, 2 mM ATP, 10 mM $MgCl_2$, 50 mM KCl, 40 µM AMdnA, and 0.9 µM AMdnC.

**Catalytic kinetic analysis of AMdnC.** A linear curve of ADP standard was established to quantitate the net production of ADP in the AMdnC reactions by HPLC analysis. ADP and ATP were separated using solvent A (30 µM $KH_2PO_4$ supplemented with 0.8 µM tetrabutylammonium phosphate (TBAP), pH 5.45) and solvent B (acetonitrile/30 µM $KH_2PO_4$, 1:1, v:v, 0.8 µM TBAP, pH 7.0). The HPLC program included the following steps: the percentage of solvent B was maintained at 10% for 0.5 min, increased to 20% over a period of 2.5 min, and then held constant for 4 min. The percentage of solvent B was then increased to 50% over a 4 min period and then held constant for 10 min. After it, the percentage of solvent B was lowered to 10% over 4 min and then held constant for 5 min. Various concentrations of AMdnA (0 to 54 µM), M1V1 (0 to 53 µM), and MdnA (0 to 42.75 µM) were processed with 1.1 µM AMdnC under optimal conditions at 37 °C and the reactions were quenched at various time points (2–40 min). For the inhibition kinetic analysis, serial concentrations of AMdnAi (0, 0.55, 1.1, 1.9, and 3.8 µM) were incubated along with AMdnA (0–37.5 µM) as described above. ADP concentrations in the reactions were determined after subtracting the peak areas of automatic ATP hydrolysis in the negative control (the same enzymatic reactions omitting substrate). For LC-based quantitation of MdnA-Δ2, we established the standard curve of MdnA and then measured the product concentrations based on the peak areas. Kinetic constants were then calculated by fitting the data to the Michaelis–Menten equation $v = V_{max}[S]/(K_m + [S])$. For competitive inhibition, the equation used was $v = V_{max}[S]/[K_m (1 + [I]/K_i) + [S]]$. All reactions were performed in triplicate, and data represented mean ± s.d.

**HPLC and high-resolution LC-MS analysis.** A Shimadzu Prominence UHPLC system (Kyoto, Japan) fitted with an Agilent Poroshell 120 EC-C18 column (2.7 µm, 3.0 × 50 mm), coupled with a PDA detector, was used for HPLC analysis. Solvent A was $H_2O$ containing 0.1% TFA and solvent B was $CH_3CN$ containing 0.1% TFA. Solvent B was applied with the following gradient: 0–1 min 10% B; 1–15 min, a linear gradient to 90% B; 18–20 min, a linear gradient to 10% B at a flow rate of 0.3 mL/min. The detection wavelengths were set at 210, 254, and 365 nm. Data from LC-HR-MS and MS/MS was obtained using a Thermo Fisher Q Exactive Focus mass spectrometer equipped with electrospray probe on Universal Ion Max API source. Acetonitrile (B)/water (A) containing 0.1% formic acid were used as mobile phases with a linear gradient program (10–90% solvent B over 15 min) to separate chemicals by the above reverse phase HPLC column at a flow rate of 0.3 mL/min. A pre-wash phase of 15 min with 10% solvent B was added at the beginning of each run, in which the elute was diverted to the waste by a diverting valve. MS1 were acquired under Full Scan mode of the Orbitrap, in which a mass range of $m/z$ 150–2000 was covered and data were collected in the positive ion mode. Fragmentation was introduced by HCD technique with optimized collision energy ranging from 20 to 30. For each selected peptide, the ion with the highest intensity was selected as the precursor ion for MS/MS analysis. Other settings for the Orbitrap scan included resolution at 15,000 and AGC target at $5 \times 10^5$. Full scan mass spectra and targeted MS/MS spectra for each of the pre-selected peptides were extracted from the raw files of the HPLC-MS/MS Experiment II using Xcalibur™ 2.1 (Thermo Scientific).

**Data availability.** All data supporting the findings of this study are available from the corresponding author upon reasonable request. Data that support the findings of this study have been deposited in the Protein Data Bank with the accession code 5IG9 and in Genbank with the accession code NC_003276.1.

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

## Acknowledgements

We thank Prof. CP Wolk (Michigan State University) for the gift of *Anabaena* sp. PCC7120. We also thank Drs. Hendrik Luesch, Valerie Paul, and Chris Xing for informative discussions and Ms. Gengnan Li and Dr. Ranjala Ratnayake for technical support. Part of this work was supported by America Cancer Society Institutional Research Grant (Y.D.) and the Department of Medicinal Chemistry at the University of Florida. Y.D. is an Air Force Office of Scientific Research Young Investigator.

## Author contributions

Y.Z., S.D.B., and Y.D. conceived the project. Y.Z. conducted all the experiments with helps from K.L., G.Y., and J.L.M.. Y.Z., S.D.B., and Y.D. analyzed the data and wrote the manuscript with helps from K.L., G.Y., and J.L.M.. All authors approved the manuscript.

## Additional information

**Competing interests:** The authors declare no competing interests.

