## [Peer Review File · Nature Communications]

Reviewers' comments:

Reviewer #1 (Remarks to the Author):

The manuscript by Zhang et al, describe characterization of a ribosomally encoded and post-translationally modified peptide biosynthetic system found in various cyanobacteria, which produces a topologically constrained macrocyclic product from a linear peptide precursor. The authors main selling point seems to be that the biosynthetic enzymes process their precursor peptides in a distributive manner and the directionality of processing depends on the particular substrate. I'm not entirely sure how this is any more "unique" or "unprecedented" than prior observation on the class II lanthipeptide syntheses, in some of which the directionality and order is contingent entirely on the substrate (i.e. ProcM, see Mukherjee, JACS 2014). Indeed, this is not unexpected given the topological constraints imposed by the primary sequence on the precursor substrate. Of course, LanMs produce only a single product, whereas the microviridins produce multiple products. Still, given the presentation in the current manuscript, it is difficult to see the impact of the discovery.

Other points:

1. The manuscript is replete with superlatives like "unique," "unprecedented", etc. when none of it is really warranted. For example on line 68, I don't see how processing multiple core peptides can be described as a "distinct" RiPP biosynthetic strategy when it extends to (at least) 5 different classes of such systems. Likewise, line 89 is not entirely correct.

2. The abstract states that (line 30), the microviridin enzymes follow a strict N-to-C directionality, unprecedented in RiPP processing enzymes. Again, this is incorrect as HalM from the lanthipeptide biosynthesis does the same (see Lee, JACS 2009).

3. The authors mention at multiple instances that the distributive processing observed in microviridins is unique among RiPP biosynthetic systems. This statement needs to be clarified as, strictly speaking, all class II lanthipeptide synthases (LanMs) are also distributive in the sense that the glutamate-addition reaction is uncoupled from the glutamate-elimination. The difference, of course, is that the LanMs process a single substrate. I get what the authors are trying to get at but the language needs to be clarified as the current description is not correct.

Reviewer #2 (Remarks to the Author):

Peptide- and protein-modifying enzymes are ubiquitous -- they are important for many biological processes, including signaling, structure, and synthesis of bioactive compounds. Here, the authors describe one such enzyme in microviridin synthesis. This is an important class of compounds, the biosynthesis of which has been characterized in previous work. Here, the advance stems from the discovery of an enzyme that makes multiple products from a single substrate, while most microviridin enzymes make a single product. By studying how the modifying enzyme processes the substrate, the authors conclude that they have found a peptide-modifying enzyme that works by an unprecedented mechanism. This is potentially a very exciting finding that would have a broad impact. However, some of the more important conclusions are not warranted in the current form of the manuscript. Improvements in that area, as well as in some underlying experiments, could make this an outstanding work of general interest.

In general, the experiments are well performed, with high-quality data shown. I don't believe, though,

that in all cases the right kind of experiments are done. Briefly, the authors make conclusions about the processing mechanism of an enzyme, but they do not describe quantitative data, such as kinetics, which are absolutely essential to make firm conclusions. A good example of where this comes up clearly is in a statement that two substrates are competitive with each other (because the rate of processing is slowed) -- this simply can't be said without testing the hypothesis quantitatively. This also comes up in comparison of mutants and reaction speeds, which are usually not valid without kinetics because of the potential for unexpected contributing factors.

Some of the conclusions can't be supported by the current evidence, including citations. Part of this might have to do with the language used. The authors state "The enzyme catalysis occurs in a distributive fashion and follows an unstrict N-to-C overall directionality, unprecedented among RiPP processing enzymes". This statement is broad enough to be useless. Which part of this is unprecedented? The combination of these elements, or each element individually? How unstrict is unstrict? The reason this comes to the fore is that I believe there are examples in the literature at least for each of these three elements individually. Therefore, a tighter definition is warranted. It may be that the authors mean this statement in terms of multiple core peptides, but it's not clear.

For example, I believe many RiPP enzymes are distributive in the sense meant by the authors. In 1999, Kelleher et al reported on the distributive nature of microcin B17 heterocyclization, and since then many other such stories have been reported.

In the sense of unstrict processing order, this is also found in ProcM, where especially in mutants the order of events can change. Perhaps the authors mean this in the multiple core peptide context? Of course, there are also N-to-C processing enzymes out there, including the microcin B17 enzyme described above.

There is a large background on microviridin kinetics and processing; this is only minimally described and referenced. A better description and reference set would be helpful to put the work in context.

Minor points:

- 1) The authors state that mutants confirm that Mg and ATP are required for catalysis; mutants cannot be used to confirm the requirement for cofactors, but rather quantitative analysis with differing amounts of those cofactors is suitable. The mutants do provide supporting evidence that Mg and ATP are required.
- 2) Evolution is mentioned in the abstract, but the meaning is not fleshed out in the body of the paper.
- 3) Line 189, since the number of dehydrations differs between substrates, relative speed of reaction is unconvincing.
- 4) It would be useful to have a simpler graphic showing the mutants used and the results. If a specialist finds it a challenge, it will be even harder for nonspecialist readers.
- 5) The cited paper about TruD is exactly on point for this manuscript. However, in making that conclusion only a single core was present, and the substrate was artificial. It therefore seems dangerous to overly generalize this conclusion. In other work on TruD and its relatives, in some of the enzymes other sites are heterocyclized first, although arguably the cited paper is done with higher quality.
- 6) Line 234, it seems obvious that the structure of a substrate drives catalysis. Maybe the meaning

could be made clearer?

7) Labels above the MS peaks should be $[M+9H]^{9+}$, etc.

Dear Reviewers:

Thank you for giving us the opportunity to revise our manuscript. It is with excitement that we submit the revised manuscript entitled “**A distributive peptide cyclase processes multiple microviridin core peptides within a single polypeptide substrate**” for publication as an Article in *Nature Communications*. We are very grateful for the very positive and constructive comments from both two reviewers, and have incorporated the suggested changes. The revised manuscript has certainly benefited from these insightful suggestions. I look forward to working with you to move this manuscript closer to the publication in *Nature Communications*.

We have responded specifically to each comment as shown below. Of note, we included one new section (“Kinetic characterization of AMdnC”) to offer quantitative understanding of the processing of multiple core peptides within a single substrate and further discussed the characterized enzyme properties in the context of both secondary and primary metabolism. Our responses were incorporated in the revised manuscript text and the revised supplementary materials. Major additions/changes in the manuscript text were marked with a red line in the right margin.

Reviewer #1:

General Comment 1:

The authors main selling point seems to be that the biosynthetic enzymes process their precursor peptides in a distributive manner and the directionality of processing depends on the particular substrate.

Author response: Thanks for the positive comment on this work! The major contribution of this work to the field of RiPPs research is **biochemical characterization of the processing of three core peptides within a single precursor peptide (AMdnA) by an ATP-grasp ligase (AMdnC)**. Processing multiple core peptides within a single substrate has so far been observed only in the production of cyanobactins, cyclotides, orbitides, ustiloxins and phomopsins. The understanding of this rare RiPPs biosynthesis strategy, however, has completely come from seminal studies of the cyanobactin biosynthetic systems. The present work significantly expanded the current knowledge by providing biochemical details of the biosynthetic logic of the second system, microviridin. We discovered that the ATP-grasp ligase AMdnC macrolactonizes all three core peptides of AMdnA in a distributive manner and demonstrates the two-levels of directionality. The distributive catalysis was retained toward multiple unnatural AMdnA analogs engineered with one to four core peptides. These processing features are different with those of the cyanobactin biosynthetic systems, indicating the mechanistic diversity in nature for the modular synthesis of RiPPs.

General Comment 2:

I'm not entirely sure how this is any more "unique" or "unprecedented" than prior observation on the class II lanthipeptide syntheses, in some of which the directionality and order is contingent entirely on the substrate (i.e. ProcM, see Mukherjee, JACS 2014). Indeed, this is not unexpected given the topological constraints imposed by the primary sequence on the precursor substrate. Of course, LanMs produce only a single product, whereas the microviridins produce multiple products. Still, given the presentation in the current manuscript, it is difficult to see the impact of the discovery.

Author response: We agree with the reviewer that the directionality of enzymatic processing of the precursor peptide is mechanistically intriguing. As indicated, a substantial amount of biochemical data accumulated over the past decade indicates that the directionality and order can be contingent entirely on the substrates as well as the enzymes (e.g., ProcM follows the C to N directionality in processing the lantipeptide substrates while HalM prefers the N to C directionality). The revised manuscript included representative RiPPs biosynthetic systems that follow the N to C, C to N and nonlinear but still ordered directionalities (lines 333-334). In this regard, our studies revealed that AMdnC follows a strict order in forming two macrolactones on each of three core peptides within AMdnA: the large ring is formed first, followed by the second small one, analogous to the N to C directionality of some processing enzymes (e.g., HalM), which typically process the substrates carrying only ONE core peptide. Importantly, AMdnC demonstrated a favored, but unstrict N to C directionality in processing THREE core peptides: its

processing can start from the first, second, or third core peptide of AMdnA, the same with AMdnA mutants. This work is the first time to report **this two-level directionality**, demonstrating the unique property of AMdnC. Accordingly, as suggested by the reviewer, we carefully interpreted our results in the context of other RiPPs biosynthetic systems. We felt that the revised manuscript properly stated the significance and novelty of AMdnC in processing AMdnA.

Minor Comment 1:

The manuscript is replete with superlatives like "unique," "unprecedented", etc. when none of it is really warranted. For example on line 68, I don't see how processing multiple core peptides can be described as a "distinct" RiPP biosynthetic strategy when it extends to (at least) 5 different classes of such systems. Likewise, line 89 is not entirely correct.

Author response: As described in our response to "General Comment 1", the modular biosynthesis has so far been reported with five families of RiPPs but only the cyanobactin biosynthesis has been characterized biochemically. Our work represented the second example with biochemical details of modular RiPPs biosynthesis and reported multiple new features of processing multiple core peptides within a single substrate. We fully agreed with the reviewer that the novelty of our work should be demonstrated in the proper context. As such, we made a number of changes in the revised manuscript to address this comment.

Minor Comment 2:

The abstract states that (line 30), the microviridin enzymes follow a strict N-to-C directionality, unprecedented in RiPP processing enzymes. Again, this is incorrect as HalM from the lanthipeptide biosynthesis does the same (see Lee, JACS 2009).

Author response: Our studies revealed that AMdnC follows an unstrict N-to-C directionality in processing three core peptides of AMdnA. As discussed above and suggested by this reviewer (e.g., HalM), such a relatively relaxed control by AMdnC is already uncommon among RiPPs processing enzymes. Importantly, when processing each single core peptide, AMdnC demonstrated a strict order in installing a large and then a small lactone. Accordingly, we rephrased the Line 30 of the previous submission as "The enzyme catalysis occurs in a distributive fashion and follows an unstrict *N-to-C* overall directionality but a strict order in installing two macrolactones on each core peptide, a unique combination among RiPPs processing enzymes".

Minor Comment 3:

The authors mention at multiple instances that the distributive processing observed in microviridins is unique among RiPP biosynthetic systems. This statement needs to be clarified as, strictly speaking, all class II lanthipeptide synthases (LanMs) are also distributive in the sense that the glutamate-addition reaction is uncoupled from the glutamate-elimination. The difference, of course, is that the LanMs process a single substrate. I get what the authors are trying to get at but the language needs to be clarified as the current description is not correct.

Author response: We are glad about the positive comment of the reviewers on the distributive processing of three core peptides as shown in this manuscript! As suggested, we carefully rephrased this discovery in the proper context as shown in this revised manuscript. Furthermore, we included a new section describing our kinetic studies to offer new mechanistic insights into the distributive catalysis of AMdnC and added two new paragraphs in the discussion to interpret our results.

Reviewer #2:

General Comment 1:

This is potentially a very exciting finding that would have a broad impact..... could make this an outstanding work of general interest.the experiments are well performed, with high-quality data shown.

Author response: Thanks for these positive comments! We were particularly very glad because these

were from the top expert in the field.

General Comment 2:

Briefly, the authors make conclusions about the processing mechanism of an enzyme, but they do not describe quantitative data, such as kinetics, which are absolutely essential to make firm conclusions.

Author response: Thanks for this constructive comment! In this work, we observed that AMdnC successfully processed AMdnA and its engineered mutants carrying one to four core peptides, and serial processing intermediates of the native and nonnative substrates were accumulated over the reaction course. These results demonstrated the distributive catalysis of AMdnC. As suggested, we performed kinetic studies of AMdnC using three peptide substrates, AMdnA, AMdnA variant M1V1 carrying only the first core peptide, and non-cognate precursor peptide MdnA from the microviridin J pathway. To circumvent the technical challenges associated with quantitating a number of potential processed intermediates (i.e., there is a maximum of 27 possible processed species, including 9 of AMdnA- Δ 3), we measured the gross reaction kinetics using the HPLC analysis to accurately quantitate the net production of ADP from ATP that accounts for all phosphorylation reactions by AMdnC, which lead to subsequent macrocyclizations. The ADP-based approach has been previously used to determine kinetic constants of other ATP-dependent enzymes (e.g., TruD in Ref. 59 and glutathione synthetase from *Arabidopsis thaliana* in Ref. 62). In addition, we designed an inactive AMdnA analog (AMdnAi) and determined it as a competitive inhibitor in the kinetic studies. Furthermore, we measured the binding affinities of selected substrates and AMdnAi with AMdnC, providing additional quantitative data to support the distributive catalysis of AMdnC. These results were included in a new section ("Kinetic characterization of AMdnC") and new supplementary figures of the revised manuscript, and we further interpreted them in a proper context as shown in "Discussion".

General Comment 3:

This also comes up in comparison of mutants and reaction speeds, which are usually not valid without kinetics because of the potential for unexpected contributing factors.

Author response: As suggested, we performed kinetic studies using three different substrates. Please refer to our responses to "General Comment 2" of Reviewer #2 for more details.

General Comment 4:

The authors state "The enzyme catalysis occurs in a distributive fashion and follows an unstrict N-to-C overall directionality, unprecedented among RiPP processing enzymes". This statement is broad enough to be useless.... a tighter definition is warranted.... many RiPP enzymes are distributive in the sense meant by the authors..... in the sense of unstrict processing order, this is also found in ProcM....

Author response: Thanks for the insightful comments! We properly addressed these comments by carefully rephrasing our conclusions and interpreting our results in the proper context, including new citations, results and discussions. Please refer to our responses to two General Comments and three Minor Comments of Reviewer #1 for more details.

General Comment 5:

There is a large background on microviridin kinetics and processing; this is only minimally described and referenced. A better description and reference set would be helpful to put the work in context.

Author response: As suggested, we cited many recent and previous microviridin papers (Refs. 4, 6, 7, 8, 11-18 and 22) and discussed the findings related to the processing. In terms of microviridin kinetics, our recent paper (*Nat. Chem. Biol.*, 2016 Nov;12: 973-979, Ref. 16) for the first time described the kinetic parameters of MdnC with MdnA as substrate. We compared our results with those in Ref. 16 and discussed and compared the two different approaches in determining the kinetic constants, which were include in the new section of this revised manuscript.

Minor Comment 1:

The authors state that mutants confirm that Mg and ATP are required for catalysis; mutants cannot be used to confirm the requirement for cofactors, but rather quantitative analysis with differing amounts of

those cofactors is suitable. The mutants do provide supporting evidence that Mg and ATP are required.

Author response: We agree with the reviewer that the designed mutants cannot be used to confirm the requirement for cofactors Mg²⁺ and ATP. As a shared mechanism among **ATP**-grasp ligases (reflected in the name of this family of enzymes), Mg²⁺ and ATP are indispensable for the enzyme reactions (e.g., glutathione synthetase from *Arabidopsis thaliana* in Ref. 62), which have been well characterized previously. Indeed, we did not detect any processed AMdnA in the AMdnC when the reaction buffer did not include either Mg²⁺ or ATP. Furthermore, we examined the divalent cation dependence of AMdnC and found the strict requirement of Mg²⁺ as showed in Supplementary Figure 6c. In this work, guided by the recent crystal structure of MdnC (Ref. 16) and sequence analysis result (Supplementary Figure 2), we designed two AMdnC mutants K165A and D280A to abolish the binding of Mg²⁺ or ATP. Expectedly, neither were active toward AMdnA (Supplementary Figure 8). Since both mutants were copurified with the same extent of impurity as AMdnC, we interpreted these results as the importance of Mg²⁺ and ATP in the AMdnC reaction and more critically the no formation of processed AMdnA by any impurity (lines 132 to 135).

Minor Comment 2:

Evolution is mentioned in the abstract, but the meaning is not fleshed out in the body of the paper.

Author response: We are very interested in probing the evolution of RiPPs biosynthesis in general and the microviridin biosynthetic systems provide excellent examples for this type of research. Indeed, our phylogenetic analysis revealed the co-evolution of cognate precursor peptides and ATP-grasp ligases. However, we did not extensively discuss the pathway evolution due to the less relevance to the main focus of the current manuscript and the length limit of *Nature Communications*. Instead, our research of pathway evolution will be disclosed in another publication later.

Minor Comment 3:

Line 189, since the number of dehydrations differs between substrates, relative speed of reaction is unconvincing.

Author response: We observed only the F3-Δ1 at the time points of 0.5 h and 2 h while the F3-Δ2 appeared only at 16 h. Accordingly, we made minor changes on the new Figure 5 to properly reflect the relatively speed in processing different core peptides by AMdnC.

Minor Comment 4:

It would be useful to have a simpler graphic showing the mutants used and the results. If a specialist finds it a challenge, it will be even harder for nonspecialist readers.

Author response: Thanks for this useful suggestion! We prepared a new figure to show the mutants used and the results, and included it as Supplementary Figure 11.

Minor Comment 5:

The cited paper about TruD is exactly on point for this manuscript. However, in making that conclusion only a single core was present, and the substrate was artificial. It therefore seems dangerous to overly generalize this conclusion. In other work on TruD and its relatives, in some of the enzymes other sites are heterocyclized first, although arguably the cited paper is done with higher quality.

Author response: We are glad that the reviewer agreed with us about the overall reaction directionality of TruD. We are also aware of other studies of TruD and its relatives. For example, the enzymes in the biosynthesis of TOMMs show an overall C to N directionality (Ref. 51), while the same is true to ThcD (a relative of TruD) in processing an artificial precursor peptide carrying two cyanobactin core peptides (Ref. 25). Nonetheless, the two-level directionality of AMdnC is truly unique and intriguing among RiPPs processing enzymes.

Minor Comment 6:

Line 234, it seems obvious that the structure of a substrate drives catalysis. Maybe the meaning could be made clearer?

Author response: As suggested, we changed the previous sentence as “Here, M1, M2 and M3 represented the first, second, and third core peptide of AMdnA (**Fig. 2c**), while V1 and V2 indicated the presence and absence of the quasi core peptide, respectively” in lines 246-248, make the naming scheme clearer. In terms of the role of the quasi core peptide in the AMdnC reaction, the results of our binding analysis (AMdnAi vs M1V1 vs AMdnA) suggested that it has a minimal effect on catalytically relevant interactions of substrate/enzyme. Indeed, AMdnC successfully processed MdnA that does not carry the quasi core peptide. Collectively, the current data did not suggest that the quasi core peptide drives the catalysis. Further studies (e.g., mutating the quasi core peptide of AMdnA) will clarify its role.

Minor Comment 7:

Labels above the MS peaks should be $[M+9H]^{9+}$, etc.

Author response: As suggested, we made the changes in corresponding figures (e.g., Figures 2, 6, and 7) in main text and supplementary information.

Thank you very much for consideration of this work. We appreciate the comments and corrections from the reviewers. We feel we have addressed the reviewer’s specific concerns regarding our manuscript and believe the manuscript is improved. Please do not hesitate to contact me if you need any additional information regarding our manuscript.

With best wishes.

Sincerely,

Yousong Ding, Ph.D.
Assistant Professor of Medicinal Chemistry

REVIEWERS' COMMENTS:

Reviewer #1 (Remarks to the Author):

The revised manuscript by Zhang and colleagues corrects many of the deficiencies of the original submission and removes most (but not all) of the unnecessary modifiers ('unique', 'novel') which were present in the initial submission. The quality of the manuscript is approaching that as would be expected of publication in a high profile journal such as Nature Communications. There are a few editorial changes that should be addressed before the manuscript may be accepted and these are detailed below:

1. The authors refer to prior work on the cyanobactin heterocyclase TruD from Koehnke et al., as illustrating C->N directionality in processing a substrate. The authors ought to make note in the Discussion section that the substrates used in that study were not physiological, as the patE substrate (uncultured *Prochloron*) is not cognate to the TruD enzyme. Moreover, the engineered substrate patE2 contained only a single cassette.
2. The sentences preceding the discussion (lines 303-305) are emphatically not correct. For processive catalysis, the concatenation of multiple core peptides as a single substrate would CHANGE the catalytic efficiency and NOT increase it. In many RiPPs, the final modifications are the slowest to be installed as the topology of the modified peptide restricts/limits further modifications.
3. I think there might be a typo on line 364 as the sentence does not make sense as written.
4. Line 461: the author contributions are missing.

Reviewer #2 (Remarks to the Author):

The authors have done an outstanding job in responding to criticism. Their manuscript really pushes the field forward in a creative way, and I'm highly supportive of publication.

I still have a criticism about claims to a uniqueness of this enzyme, as stated on lines 176-180 (and similarly in lines 334-353, although stated better there). This enzyme is not unique or unusual in the manner stated. In fact, it may be similar, although other enzymes don't have the quality of presentation as in the current work. I don't think any similarity with other enzymes detracts from the paper at all, and in fact makes it stronger, more interesting, and more general. It's part of what makes the experimental approach of this paper so good.

The main point I was trying to make is that the cyanobactin heterocyclase is distributive. Although much work supports this, there is direct phrasing in one of the previous cyanobactin papers that states this (McIntosh *ChemBioChem* 2010): "Quite clearly then, the precursor peptide leaves the enzyme after the second heterocyclization and before the third heterocyclization." "These experiments are most consistent with the idea that the substrate can dissociate from the enzyme between heterocyclizations"... Again, there is also later experimental work with multi-cassette precursors that also supports this as well.

I don't think the authors need to change much about the paper to address this, and they don't need to cite anything further. However, some phrasing needs to change, unless they can provide evidence overturning those observations. Examples:

Lines 79-80 and 176: The authors cite reference 32 in support of this idea. Reference 32 uses an artificial, single-cassette substrate. The results are virtually identical to the within-cassette results shown in this paper, e.g. on lines 198, 206, and 219. What reference 32 doesn't describe is the directionality in multiple cassettes, which is what the authors are trying to compare with their work here.

*Minor further things to consider.

Line 67: Do the authors mean Fig 1b?

Line 205: topological? I think topical may not be the right word.

(A final note: Line 267: Although I like the way the kinetics are done, almost all E. coli preps of precursor peptides or enzymes obtained in the manner described contain other active enzymes, such as AMP kinase, which can skew results. ATP using enzymes could include chaperones. So I think it might be dangerous to make the conclusion in line 283-284. Up to authors.)

Dear Reviewers:

Thank you for giving us the opportunity to revise our manuscript (NCOMMS-17-11055A). It is with excitement that we submit the final version of this manuscript entitled “**A distributive peptide cyclase processes multiple microviridin core peptides within a single polypeptide substrate**” for publication as an Article in *Nature Communications*. We are very grateful for the very positive and constructive comments from both two reviewers, and have incorporated the suggested changes. The revised manuscript has certainly benefited from these insightful suggestions. I look forward to working with you to move this manuscript for the publication in *Nature Communications*.

We have responded specifically to each comment as shown below. Our responses were incorporated in the revised manuscript text, shown in the mode of track changes, and the revised supplementary materials.

Reviewer #1:

General Comment 1:

The revised manuscript by Zhang and colleagues corrects many of the deficiencies of the original submission and removes most (but not all) of the unnecessary modifiers ('unique', 'novel') which were present in the initial submission. The quality of the manuscript is approaching that as would be expected of publication in a high profile journal such as *Nature Communications*.

Author response: Thanks for the positive comments on the previous version of our manuscript! Our work has benefited greatly from your comments and suggestions.

General Comment 2:

The authors refer to prior work on the cyanobactin heterocyclase TruD from Koehnke et al., as illustrating C->N directionality in processing a substrate. The authors ought to make note in the Discussion section that the substrates used in that study were not physiological, as the patE substrate (uncultured *Prochloron*) is not cognate to the TruD enzyme. Moreover, the engineered substrate patE2 contained only a single cassette.

Author response: We agree with the reviewer that catalytic performance of TruD can be influenced by the substrates used in the enzyme reactions. Accordingly, we carefully rephrased the sentences about the processivity and directionality of cyanobactin heterocyclase in this version of manuscript (e.g., lines 77-80, lines 192-194, lines 213-214, lines 359-360 and lines 371-373), which now states the catalytic properties of TruD more accurately.

General Comment 3:

The sentences preceding the discussion (lines 303-305) are emphatically not correct. For processive catalysis, the concatenation of multiple core peptides as a single substrate would CHANGE the catalytic efficiency and NOT increase it. In many RiPPs, the final modifications are the slowest to be installed as the topology of the modified peptide restricts/limits further modifications.

Author response: Thanks for this comment! In other RiPP systems with a single core peptide, multiple enzymes may be needed to synthesize the final products, and the reaction by the last enzyme is typically the slowest. By contrast, in this work, we determined the overall reaction rates of a single enzyme AMdnC in catalyzing multiple reactions on one to more microviridin core peptides within a single substrate. This system is thus principally different with the majority of other characterized RiPP systems, and it remains less clear whether the catalytic kinetics of AMdnC follows the same rule as others. Nonetheless, we rephrased this sentence as “For processive catalysis, the concatenation of multiple modification sites within a single substrate could increase the catalytic efficiency as reducing the search dimension of enzyme to one (*N* to *C* or *C* to *M*), as shown with the restriction enzyme *EcoRI* in a previous report⁴⁵.”

General Comment 4:

I think there might be a typo on line 364 as the sentence does not make sense as written.

Author response: To avoid any potential confusion, we changed this sentence to “On the other hand, the apparent rates (k_{cat}) of ATP hydrolysis in the AMdnC reactions were 20-60 times faster than those determined by the HPLC method (apparent k_{cat} values varied from 0.47 min⁻¹ to 0.7 min⁻¹)”.

General Comment 5:

Line 461: the author contributions are missing.

Author response: The statement of author contributions is now included.

Reviewer #2:

General Comment 1:

The authors have done an outstanding job in responding to criticism. Their manuscript really pushes the field forward in a creative way, and I’m highly supportive of publication.

Author response: We greatly appreciate all positive comments from the reviewer!

General Comment 2:

I still have a criticism about claims to a uniqueness of this enzyme, as stated on lines 176-180 (and similarly in lines 334-353, although stated better there). This enzyme is not unique or unusual in the manner stated. In fact, it may be similar, although other enzymes don’t have the quality of presentation as in the current work. I don’t think any similarity with other enzymes detracts from the paper all, and in fact makes it stronger, more interesting, and more general. It’s part of what makes the experimental approach of this paper so good.

Author response: Thanks for this valuable criticism! The highly similar comment was provided by the reviewer 1. As shown in our response to “General Comment 2” of the reviewer 1, we followed suggestions of both reviewers to describe the catalytic features of cyanobactin heterocycles in a more precise manner, which truly makes our claim to the uniqueness of AMdnC in this revised manuscript to be more justified. We are very glad about the positive comments on the approaches used in this work.

General Comment 3:

The main point I was trying to make is that the cyanobactin heterocyclase is distributive. Although much work supports this, there is direct phrasing in one of the previous cyanobactin papers that states this (McIntosh ChemBioChem 2010): “Quite clearly then, the precursor peptide leaves the enzyme after the second heterocyclization and before the third heterocyclization.” “These experiments are most consistent with the idea that the substrate can dissociate from the enzyme between heterocyclizations”... Again, there is also later experimental work with multi-cassette precursors that also supports this as well.

Author response: We agreed with this reviewer that cyanobactin heterocyclase can be distributive and its catalytic features are seemingly influenced by the nature of (natural or engineered) cyanobactin substrates. In this final revision, we followed the suggestion of this reviewer as well as the reviewer 1 to properly describe the catalytic performances of cyanobactin heterocyclase, as detailed in our response to “General Comment 2” of the reviewer 1.

General Comment 4:

I don’t think the authors need to change much about the paper to address this, and they don’t need to cite anything further. However, some phrasing needs to change.... Lines 79-80 and 176: The authors cite reference 32 in support of this idea. Reference 32 uses an artificial, single-cassette substrate. The results are virtually identical to the within-cassette results shown in this paper, e.g. on lines 198, 206, and 219. What reference 32 doesn’t describe is the directionality in multiple cassettes, which is what the authors are trying to compare with their work here.

Author response: Thanks for the insightful comments! We properly addressed this comment by rephrasing many sentences about cyanobactin heterocyclase and sometimes included additional, proper references (e.g., ref 59, McIntosh, ChemBioChem 2010) along with ref32.

Minor Comment 1:

Line 67: Do the authors mean Fig 1b?

Author response: According to the policies of Nature Communications, we moved Fig. 1b of the original Figure 1 as Fig. 2a in the new Figure 2. Therefore, there is no need to cite a figure in the corresponding sentence.

Minor Comment 2:

Line 205: topological? I think topical may not be the right word.

Author response: Thanks for the suggestion! We replaced all “topical” with “topological” in the revision.

Minor Comment 3:

A final note: Line 267: Although I like the way the kinetics are done, almost all E. coli preps of precursor peptides or enzymes obtained in the manner described contain other active enzymes, such as AMP kinase, which can skew results. ATP using enzymes could include chaperones. So I think it might be dangerous to make the conclusion in line 283-284. Up to authors.

Author response: Initially, we shared the same concern as this reviewer. Accordingly, we subtracted the ADP produced by automatic ATP hydrolysis or any ATP using enzymes co-purified with AMdnC in the control reactions that were the same as enzymatic reactions except no substrate were used. Furthermore, we observed the no-to-minimal net production of ADP in the AMdnC reaction containing AMdnAi. AMdnAi is an inactive substrate of AMdnC, and showed a similar level of purity to AMdnA as shown in SDS-PAGE analysis (Supplementary Fig. 3). Collectively, these results suggested that the net production of ADP in the AMdnC reaction is driven by AMdnC.

Thank you very much for consideration of this work. We appreciate the comments and corrections from the reviewers. We feel we have addressed the reviewer’s specific concerns regarding our manuscript and believe the final version of this manuscript is improved. Please do not hesitate to contact me if you need any additional information regarding our manuscript.

With best wishes.

Sincerely,

Yousong Ding, Ph.D.
Assistant Professor of Medicinal Chemistry